# Epigenetic Modifications of the Liver Tumor Cell Line HepG2 Increase Their Drug Metabolic Capacity

**DOI:** 10.3390/ijms20020347

**Published:** 2019-01-16

**Authors:** Marc Ruoß, Georg Damm, Massoud Vosough, Lisa Ehret, Carl Grom-Baumgarten, Martin Petkov, Silvio Naddalin, Ruth Ladurner, Daniel Seehofer, Andreas Nussler, Sahar Sajadian

**Affiliations:** 1Siegfried Weller Institute, BG Trauma Clinic, Eberhard Karls University Tübingen, 72076 Tübingen, Germany; m.ruoss@hotmail.de (M.R.); lisa.ehret@gmx.de (L.E.); carlgrom-baumgarten@web.de (C.G.-B.); petkov.martin12@gmail.com (M.P.); sahar.sajadian@gmail.com (S.S.); 2Department of Hepatobiliary Surgery and Visceral Transplantation, University of Leipzig, 04103 Leipzig, Germany; georg.damm@medizin.uni-leipzig.de (G.D.); daniel.seehofer@medizin.uni-leipzig.de (D.S.); 3Royan Institute for Stem Cell Biology and Technology, Department of Stem Cells and Developmental Biology, Tehran 16635-148, Iran; masvos@yahoo.com; 4Department of General, Visceral and Transplant Surgery, University Hospital Tübingen, 72076 Tübingen, Germany; silvio.nadalin@med.uni-tuebingen.de (S.N.); ruth.ladurner@med.uni-tuebingen.de (R.L.)

**Keywords:** tumor cells, epigenetic reprogramming, drug metabolism, hepatoma cell lines, primary human hepatocytes

## Abstract

Although human liver tumor cells have reduced metabolic functions as compared to primary human hepatocytes (PHH) they are widely used for pre-screening tests of drug metabolism and toxicity. The aim of the present study was to modify liver cancer cell lines in order to improve their drug-metabolizing activities towards PHH. It is well-known that epigenetics is strongly modified in tumor cells and that epigenetic regulators influence the expression and function of Cytochrome P450 (CYP) enzymes through altering crucial transcription factors responsible for drug-metabolizing enzymes. Therefore, we screened the epigenetic status of four different liver cancer cell lines (Huh7, HLE, HepG2 and AKN-1) which were reported to have metabolizing drug activities. Our results showed that HepG2 cells demonstrated the highest similarity compared to PHH. Thus, we modified the epigenetic status of HepG2 cells towards ‘normal’ liver cells by 5-Azacytidine (5-AZA) and Vitamin C exposure. Then, mRNA expression of Epithelial-mesenchymal transition (EMT) marker SNAIL and CYP enzymes were measured by PCR and determinate specific drug metabolites, associated with CYP enzymes by LC/MS. Our results demonstrated an epigenetic shift in HepG2 cells towards PHH after exposure to 5-AZA and Vitamin C which resulted in a higher expression and activity of specific drug metabolizing CYP enzymes. Finally, we observed that 5-AZA and Vitamin C led to an increased expression of Hepatocyte nuclear factor 4α (HNF4α) and E-Cadherin and a significant down regulation of Snail1 (SNAIL), the key transcriptional repressor of E-Cadherin. Our study shows, that certain phase I genes and their enzyme activities are increased by epigenetic modification in HepG2 cells with a concomitant reduction of EMT marker gene SNAIL. The enhancing of liver specific functions in hepatoma cells using epigenetic modifiers opens new opportunities for the usage of cell lines as a potential liver in vitro model for drug testing and development.

## 1. Introduction

Drug metabolism is understood to mean the biochemical process which describes the modification of drugs, which has the purpose to inactivate a substance and excrete it from the body. Changes in the gene expression of enzymes specialized in drug metabolism may result in altered metabolism of the respective substance [1]. In recent years, it has been shown in many studies that the epigenetic regulation of drug-metabolizing enzymes is an important mechanism here. Epigenetic regulation occurs in three stages: 1. nucleosome positioning 2. histone modification 3. DNA methylation [2]. Recent studies have revealed that epigenetic factors regulate the expression of drug-metabolizing enzymes and the drug transporter [1,3]. Over recent years, modifying the epigenetic status of genes responsible for increasing Cytochrome P450 (CYP) enzyme activities attracted more attention [4]. Epigenetic modifications are closely linked to the so-called Epithelial-mesenchymal transition (EMT) [5]. The EMT process not only has a pronounced influence on cell metabolism but also plays an important role in the degree of differentiation of the cells, embryogenesis, liver fibrosis and metastasis of cancer cells [6,7]. EMT additionally enables an increased immunosuppression and drug resistance, which correlates with the epigenetic status of the tumor cell [8]. EMT in hepatocytes is associated with an overexpression of Snail1 (SNAIL), which downregulates the key epithelial marker gene E-Cadherin and other hepatic differentiation key factors such as HNF4α [8,9]. It was shown that the most important hepatic genes like Hepatocyte nuclear factor 4α (HNF4α) are influenced by epigenetic regulators such as HDACi (Histone deacetylase inhibitors) and DNMTi (DNA methyltransferase inhibitors) [10]. Epigenetic modification promotes growth arrest and up-regulates the expression of the hepatic key regulator gene *HNF4α* in various hepatoma cells which induces increased CYP expression and Albumin production [11]. Therefore, modifying and triggering the epigenetic state of hepatoma cell lines may change the expression of genes responsible for CYP activities. Recently, we have demonstrated that the cytidine analogue 5-Azacytidine (5-AZA) and Vitamin C reduce the gene and protein expression of SNAIL in the Hepatocellular carcinoma (HCC) cell lines Huh7 and HLE [12]. Various studies focused on the effect of DNMTi such as 5-AZA and 5-Aza -2′-deoxycytidine (5-AZA-dC) on the expression of crucial phase I and II biotransformation genes and some of them suggested improvement of the CYP3A4, CYP3A7, CYP1B, UDP-Glucuronosyltransferase-2B15 and Glutathione S-transferase P1 gene expression [10]. Additionally, it is known that insulin contributes to the preservation of hepatocytes morphology and the glucocorticoids support the maintenance of differentiation which is crucial for the function of CYPs [13,14]. 

Therefore, the overall aim of this study was to improve the metabolic function of liver tumor cell lines towards primary human hepatocytes (PHH) by modifying their epigenetic status. First, we have examined the expression level of epigenetic modifying enzymes in four hepatoma cell lines (HepG2, Huh7, HLE and AKN1) that have been reported having less liver metabolic functions [15,16] than freshly isolated PHH. The cell line HepG2 shows the highest similarity in its epigenetic profile compared to PHH was used for further testing. Here we have shown how the expression levels of metabolic related genes and enzyme activities change after treatment with Vitamin C in combination with 5-AZA. Moreover, we investigated the influence of these changes on the EMT and the hepatic key regulator genes. Finally, we tested the effect of classical media supplements from hepatocyte culture media, such as insulin and hydrocortisone on CYP activity in hepatoma cell lines, that are usually not included in the maintenance medium of these hepatoma cell lines [15] may further improve the hepatic metabolic function of liver tumor cells. 

## 2. Results

### 2.1. The Regulation of the Epigenetic Enzymes in HepG2 is Most Closely Comparable to the Expression of Primary Human Hepatocytes

For epigenetic characterization of the investigated liver cell lines, we investigated the expression of chromatin remodeling enzymes and compared to the results to PHH. For the characterization, we used the Human Epigenetic Chromatin Modification Enzymes PCR Array from QIAGEN. The analysis of the real-time PCR results revealed that each individual tumor cell line showed an individual profile of chromatin-modifying genes compared to human hepatocytes (Figure 1, Appendix A). The largest differences in the pattern of chromatin modifying proteins were seen in the Huh7 cells compared to PHH, whereas HepG2 cells showed the highest similarity to PHH among all tested liver tumor cell lines. Therefore, in the further course of the study we have focused on the usage of the cell line HepG2. Then, we tested the possibility whether or not 5-AZA and/or Vitamin C incubation reduces existing epigenetic differences compared to PHH and whether these epigenetic modifications result in an increase of the metabolic function of the HepG2 cell line.

### 2.2. Treatment of HepG2 with Epigenetic Modifying Compounds Revealed a Positive Impact on the Expression of Genes From Xenobiotic Metabolism

Since the HepG2 cell line showed epigenetically the highest similarity to PHH, this cell line was chosen for further investigations. Exposure of HepG2 cells to 5-AZA and Vitamin C resulted in dramatic changes in the expression of chromatin-modifying enzymes (Figure 2) that are known for their significant role in the maintenance of metabolic properties of hepatocytes. The stimulation with Vitamin C alone also leads to changes, but to a lesser extent, and therefore the combination of 5-AZA plus Vitamin C was used for the further experiments. The expression of all 84 measured genes can be found in the Appendix A. As shown in Figure 2 the treatment of the cells with 5-AZA plus Vitamin C has an effect on the expression level of most of the investigated genes.

### 2.3. Stimulation of HepG2 Cells with Epigenetic Modifying Compounds Result in Changes in Gene Expression of Epigenetic Modifying Enzymes

The results of the Chromatin Modification Array have been validated by qPCR. Additionally, it was tested, whether insulin and hydrocortisone show a positive effect on the expression of the measured genes. These two substances were chosen because it is well known that they play an important role in hepatic differentiation and maintenance of the function of PHH in vitro.

As shown in Figure 3 The treatment of cells with 5-AZA and Vitamin C with/without insulin and hydrocortisone changed the expression of all measured genes compared to non-treated one. The treatment decreased the level of the mRNA expression of histone deacetylase (*HDAC) 1 and 2*, which reached the level of PHH. 

In contrast, lysine demethylase 6B (*KDM6B*) gene expression increased significantly by 5-AZA and Vitamin C, although the expression level didn’t reach the level of PHH. The addition of insulin and hydrocortisone to the treatment showed a further improvement for lysine demethylase 4C (*KDM4C*) compared to treatment with only 5-AZA and Vitamin C stimulated cells. All other investigated genes are not affected by additional treatment. 

### 2.4. Stimulation of HepG2 Cells with 5-AZA Plus Vitamin C Led to the Downregulation of the EMT Marker Gene SNAIL, an Increase of Epithelial Marker Genes and the Hepatic Key Regulator HNF4α

In our previous study, we have shown, treating hepatoma cell lines with 5-AZA and Vitamin C decreased *SNAIL* expression which is associated with an increased expression of the epithelial marker gene *E-Cadherin* [12]. As shown in Figure 4, we found similar results in the hepatoblastoma cell line HepG2. Interestingly, adding insulin and hydrocortisone to the medium enhance the expression *E-Cadherin* more than threefold compared to PHH. The expression of the epithelial gene Cytokeratin 18 (*CK18*) was increased by the treatment with 5-AZA plus Vitamin C and almost reached the level of PHH by further addition of insulin and hydrocortisone. Furthermore, we observed a slight change in *HNF4α* gene expression after incubation of HepG2 cells with 5-AZA plus Vitamin C, which was not further increased after insulin and hydrocortisone supplementation. Although the treatment with 5-AZA and Vitamin C significantly increased *HNF4α* expression, it was still less than 10% of *HNF4α* expression observed in PHH.

### 2.5. Treatment of HepG2 Cells with 5-AZA Plus Vitamin C in Combination with Insulin and Hydrocortisone Resulted in an Increased CYP450 Gene Expression and Enzyme Activity

The next aim was to verify if the above described molecular changes after AZA plus Vitamin C incubation had any positive impact on CYP450 gene expression and enzyme activity. Additionally, we wanted to test if the supplementation with insulin and hydrocortisone leads to a further increase of gene expression and activity of the CYP450 enzymes. Therefore, we investigated the gene expression of the following CYPs: *CYP1A2*, *CYP3A4* and *CYP2C9*. As depicted from Figure 5 we observed an upregulation of all three CYP genes after incubation of HepG2 cells with 5-AZA, Vitamin C, insulin and hydrocortisone. It is noteworthy that incubation of HepG2 with 5-AZA plus Vitamin C alone did not result in any *CYP3A4* upregulation, but supplementation of insulin and hydrocortisone did. Nevertheless, although we observed an increased CYP gene expression following 5-AZA, Vitamin C, insulin plus hydrocortisone in HepG2 cells, the gene expression of the cell line is still much lower than in freshly isolated human hepatocytes.

Next, we measured the CYP450 enzyme activity after incubation with epigenetic modifiers. Using LC /MS technology the metabolization of specific drugs was carried out [17]. The results of the activity measurements (Figure 6) showed only a modest increase of CYP3A4, CYP2D6 and CYP2C9 enzyme activities in HepG2 cells stimulated with 5-AZA and Vitamin C. Additional supplementation of insulin and hydrocortisone to the medium resulted in a further modest increase. However, the shown increases are not significant and are still far from the activities of PHH. For CYP1A2 no activity could be measured in HepG2. 

The result of our study indicated that 5-AZA and Vitamin C change the expression of some epigenetic markers that affect some crucial genes in terms of hepatic functions. These alterations are associated with an increase in the expression of epithelial marker genes such as *HNF4α* and *E-Cadherin* as well as a decrease of the EMT marker gene *SNAIL*. However, there is no significant increase in the expression or activity of drug metabolizing enzymes. So as to identify chromatin-modifying genes that may be associated with low gene expression and activity of the CYP enzymes we looked again at the chromatin array data to see if there are any genes that are de-regulated in hepatoma cell lines but have not yet been linked to the metabolic activity of PHH. As shown in Figure 7, these criteria apply to the genes *SUV39H1*, *SMYD3*, *SETDB2*, *ESCO2*, *AURK A* and *AURK B*. As the results show, these genes are massively de-regulated in all tested cell lines. However, the gene expression profile of the cell line HepG2 is still the closest to PHH, compared to the other tested cell lines. A change in the expression of these genes can also be found, to a lesser extent, in de-differentiated PHH at day 7 after plating. Stimulation with 5-AZA and Vitamin C has no significant effect on the expression of these genes in HepG2. The expression of all 84 measured genes after treatment with Vitamin C alone or with 5-AZA plus Vitamin C can be found in the Appendix A.

## 3. Discussion

The culture of PHH, as well as hepatoma cell lines, are two of the most common in vitro liver models to evaluate drug metabolism and hepatotoxicity. Unfortunately, cultured primary hepatocytes lose their drug metabolic capacity rapidly in culture and they have large batch-to-batch variations. In contrast, hepatoma cell lines have an unlimited life span and they consist of a more stable phenotype than primary hepatocytes. Additionally, hepatoma cell lines are constantly available, but they show low CYP activities [18]. One of the main causes of the reduced metabolic activity of tumor cell lines are epigenetic changes, which might be associated with a so-called EMT. Recently we could clearly show that the treatment with a combination of 5-AZA and Vitamin C not only leads to a stop of proliferation but also results in a positive effect on EMT in HCC cell lines [12]. Moreover, we have shown that epigenetic changes of Ad-MSCs increased the metabolic capacity of differentiated hepatocyte-like cells. An increased gene expression and activity of CYP 450 (CYP1A2, CYP2E1, CYP3A4/7, CYP2D6, CYP2B6) enzymes were achieved by treatment with the epigenetic modifiers 5-AZA plus Vitamin C [19]. The aim of this study was to investigate whether epigenetic modifications in human hepatoma cell lines used to study drug metabolism would increase their metabolic capacity. 

Our results clearly show that all of the tested liver tumor cell lines that are commonly used for drug metabolism show large differences in their epigenetic profile compared to PHH. Among the tested cell lines, it is striking that the hepatoblastoma cell line HepG2 differ significantly in the expression of epigenetic modifying enzymes from other tested HCC cell lines. In addition, our results showed that HepG2 cell line is most comparable in their epigenetic profile to PHH, which was the reason for continuing our research for this manuscript with this cell line. Despite the highest similarity of epigenetic profile between HepG2 and PHH, there are still significant differences. For example, the expression of *HDAC2* is up-regulated in HepG2 compared to PHH, this upregulation was also found in HCC tissue [20]. The overexpression of this HDAC correlates with the dedifferentiation state and increased proliferative activity of tumor cells [20]. Our results also demonstrated that *PRMT1* is upregulated in HepG2 cells. This correlates which the findings of Gou et al. who found an upregulation of PRMT1 in several liver cancer cells as well as in HCC tissue. Furthermore, the authors have shown that knockdown of PRMT1 reverses EMT in HCC cell lines underlining the important role between PRMT1 and EMT and proliferation [21]. The downregulation of *KDM4C* which we have found in untreated HepG2 cells is also an important finding since it has diverse targets in oncogenic or tumor suppressor functions [22]. The reduction of *KDM4C* gene expression in HepG2 most likely causes the methylation of E-Cadherin promoter [23]. Additionally, we found a low basic *KDM6B* gene expression in HepG2 cells compared to PHH that is in line with findings in various cancers, including liver carcinoma [24]. KDM6B is of particular importance because its function as a histone demethylase that specifically demethylates Lys-27 of histone H3 tri- and dimethyl (me3/2) [25] and therefore explains the low basal *HNF4α* levels observed in HepG2 cells. The hepatic key regulator HNF4α showed only minimal levels of H3K27me3 enrichment at these promoters in primary hepatocytes. In other cells, removal of H3K27me3 from the HNF4α promoter resulted in transcriptional activation and expression of liver enrichment transcription factors [26]. A degradation of this methylation in HepG2 could be associated with an increased expression of *HNF4α* which can be attributed to the activity of KDM6B [25]. The literature data for the expression of *HDAC1* and *8* as well as *CARM1* indicating that these genes also have influence on tumorigenesis, EMT and the hepatic function [4,10,27,28] but they are not de-regulated in HepG2 however, we have found a de-regulation in the other tested HCC cell lines, which was another reason for using the cell line HepG2 for the further experiments.

In the second part of our study, we investigated whether the epigenetic modifications in HepG2 could be altered by treatment with 5-AZA plus Vitamin C towards PHH. Our results clearly show that HepG2 treatment with 5-AZA plus Vitamin C led to a significant reduction of the EMT marker gene *SNAIL* and a significant increase of *E-Cadherin* gene expression, which is in line with other human cancer cell lines [29]. Interestingly, additional stimulation of cells with insulin plus hydrocortisone further enhances E-Cadherin expression which confirms observations by Zhaeentan et al. showing that glucocorticoids increased E-cadherin gene expression [30]. The increased expression of *HNF4α* which results from the treatment with 5-AZA plus Vitamin C could be explained by the downregulation of *SNAIL* [9]. Moreover, HNF4α is essential in liver development and differentiation, lipid homeostasis, bile acid synthesis, as well as the expression of phase I, II, and III drug metabolizing genes. Aberrations in HNF4α functionality are known from the development of severe cirrhotic livers, alcoholic liver disease, tumor necrosis factor-α-induced hepatotoxicity, and hepatocellular carcinoma where HNF4α has an anti-proliferative effect and serves as a tumor suppressor [31]. We have also found a strong increase of *CK18* gene expression following the incubation with 5-AZA, Vitamin C, insulin and hydrocortisone. CK18 is one of the crucial epithelial markers and was expressed after treatment comparable to PHH [32]. Since treatment with 5-AZA plus Vitamin C results in a positive change of the expression of epigenetic modifiers and in downregulation of the EMT marker gene SNAIL along with an increase of epithelial gene expression markers, we hypothesized an increased metabolic activity of CYP enzymes. However, our results only show a slight, but not significant, increase in CYP gene expression and activity compared to PHH. This is in line with data of Dannenberg et al. showing especially for *CYP3A4* a 1.8-fold increase after the treatment with 5-aza-dC in HepG2 cells. Moreover, this study also demonstrated that additional treatment with the HDAC inhibitor TSA did not contribute to an improvement in metabolic activity [33]. However, based on our earlier results, the question arises why a positive change on the expression and activity of CYP enzymes can be achieved in Ad-MSCs, which were differentiated under treatment with 5-AZA [19], but not in hepatoma cells. One possible explanation could be given by the results of Weng et al. showing that the profile of epigenetically modifying enzymes from embryonic stem cells was much closer to the profile of PHH [34]. Most striking was the de-regulation of the DNA methyltransferase 3B (*DNMT3B*) gene and the depletion of DNMT3B from soluble fraction after 5-aza-dC treatment, resulted in an enhanced differentiation of Ad-MSCs [35]. Although HepG2 cells have the closest epigenetic profile to PHH from all investigated liver cell lines, which was even further improved after 5- AZA and Vitamin C treatment, the overall improvement of metabolic activity in HepG2 cells remained bleak. Therefore, for a substantial improvement of the metabolic profile in hepatoma cell lines, further studies are needed to identify additional key regulators. Nevertheless, our results are important because they clearly show that also other as the known epigenetic modifying enzymes such as HDACs or DNMT [10] influence the metabolic profile of hepatoma cell lines. The function of these genes with respect to drug metabolism was until now unknown. In summary, our findings support earlier results suggesting that drug induced epigenetic alteration of HCCs might be useful, for example, with combination therapy with 5-AZA and Vitamin C, as it increases expression of epithelial genes and as described by Sajadian et al. to stop or at least reduce tumor proliferation and reverse EMT [12]. However, as shown in our data presented here, additional inhibition of genes such as *Aurora kinases A* and *B* or *ESCO2*, which are still strongly de-regulated, might be useful to increase the expression of known hepatic functional genes such as *HNF4α* to the level of hepatocytes [9]. It may also be possible to increase other genes such as *KDM4C* or *KDM6B* which, as already described, also have an influence on the expression of hepatic genes on the expression level of hepatocytes [22,25]. Overall, this could also be accompanied by an increase in the expression of the CYP enzymes, which would then make the epigenetic modified cells to an interesting in vitro test system for the development of new drugs.

## 4. Materials and Methods

### 4.1. Tissue Samples

Liver cells were isolated from macroscopically tumor free tissue that remained from resected human liver of patients with primary or secondary liver tumors or benign local liver tissue. The liver capsules, were collected at the clinic of General, Visceral and Transplant Surgery (Tübingen) and the clinic of Hepatobiliary Surgery and Visceral Transplantation (Leipzig). Informed consent of the patients for the use of tissue for research purposes was obtained according to the ethical guidelines of the Ethic commission of the medical faculty of the University of Tübingen, Tübingen, Germany project number: 368/2012BO2 (02 August 2012) and the Ethic commission of the medical faculty of the University of Leipzig, Leipzig, Germany project number: 177/16-IK (12 July 2016).

### 4.2. Isolation of Primary Human Hepatocytes

PHH were isolated using a two-step EDTA/collagenase perfusion technique as described elsewhere [36,37]. The cells were shipped overnight in ChillProtect Plus solution (Biochrom, Berlin Germany). Upon arrival, the cell number and viability were determined in a Neubauer counting chamber using Trypan blue. If the viability of the isolated cells as determined by Trypan blue staining was below 70%, density gradient centrifugation was performed to remove non-viable cells. In brief, the cell pellet containing the PHH fraction was subjected on a 25% Percoll solution (total density: 1.0675 g/L) and centrifuged at 1250 × g, 20 min, 4 °C without brake. The resulting PHH fraction was washed with PBS (w/o Mg^2+^, Ca^2+^) and re-suspended in PHH culture medium (Williams Medium E supplemented with 10% Fetal bovine serum, 100 U/mL Penicillin, 0.1 mg/mL Streptomycin, 15 mM HEPES, 1 mM Glutamine, 1 mM Sodium pyruvate, 1 mM human Insulin, 0.8 µg/mL Hydrocortisone and 1% Nonessential amino acids). 

### 4.3. Culture of Primary Cells and Cell Lines

#### 4.3.1. Primary Cells

For adherence, culture dishes were coated with rat tail collagen as described elsewhere [9]. Cells were seeded in a density of 1.5 × 10^5^ cells/cm^2^ and cultured in PHH culture media.

#### 4.3.2. Cell Lines

The common liver cancer cell lines HepG2, Huh7 and HLE as well as the cell line AKN1 were used in this study. HepG2 and HLE cell line were purchased from ATCC, Huh7 from JCRB (Japanese Collection of Research Bioresources Cell Bank, Osaka, Japan). The cell line AKN1 was isolated and developed as described elsewhere [16]. The cell lines were cultured as described [38]. An overview of the cell lines used can be found in Table 1.

The absence of mycoplasma contamination was regularly confirmed using a commercially available test kit (Mycroalert Detection Kit Cat. No: LT07, Lonza, Basel, Switzerland). The HCC cell lines were plated on day 0 in a density of 8 × 10^3^ cells/cm^2^. On day 1, the medium was removed and the cells were washed once with PBS. Subsequently, the cells were incubated for 48 h in a stimulation medium. The stimulation medium contained 5-AZA and Vitamin C (see Condition 1). Additional incubation with hydrocortisone and insulin was achieved by adding these compounds to the medium containing 5-AZA, and Vitamin C (see Condition 2) at day 2. The schedule of the stimulation is shown below in Table 2.

### 4.4. Epigenetic Modification Array

To investigate the variation of chromatin modification among HCC cell lines and PHH, a Chromatin Modification array was used. Huh7, HepG2, HLE, AKN1 were investigated and compared to PHH. RNA of 5 different passage numbers of HCC cell lines and 5 different donors of PHH were isolated. RNA of different passage numbers/different PHH donors were pooled. The pooled RNA was purified with Rneasy Lipid Tissue Mini Kit according to manufacturer’s instruction (QIAGEN, Germantown, MD, USA).

RT^2^ First strand synthesis kit was used in order to eliminate genomic DNA contamination and reverse transcription was performed according to manufacturer’s protocol (QIAGEN). RT^2^ SYBR Green mastermix (QIAGEN) was used for performing real-time PCR for RT^2^ PCR array according to manufacturer’s protocol, in brief: denaturation for 10 min at 95 °C, amplification with 40 cycles and 15 s at 95 °C, 1 min at 60 °C and 15 s at 72 °C (Step One Plus^TM^ Real -Time PCR System, (Life Technologies, Carlsbad, CA, USA). The Human Epigenetic Chromatin Modification Enzymes RT² Profiler™ PCR Array (Cat. No: 33231 PAHS-085ZA, QIAGEN). The array measures the expression of 84 genes which encode for epigenetic key enzymes responsible for modifying histones as well as the packing of DNA strands into chromosomes and are consequently linked to the regulation of gene expression. The array includes genes acting as DNA methyltransferases, enzymes that catalyze histone acetylation, methylation, phosphorylation, ubiquitination and histone deacetylases and demethylases.

### 4.5. cDNA Synthesis and RT-PCR 

Real-time RT-PCR for detection of mRNA expression was performed as described previously [38]. In brief, for the mRNA expression studies, total RNA was extracted using TriFast reagent (Peqlab, Erlangen, Germany). Complementary DNA (cDNA) was synthesized by First Strand cDNA Synthesis Kit (Thermo Scientific, Waltham, MA, USA). For quantitative real-time PCR (qRT-PCR), 40 ng of template cDNA was used for the expression level of each target gene (Gene sequences of the primers which were used can be found in Table 3 using SYBR Green qPCR (Thermo Scientific) and the Step One Plus® Real-Time PCR System Kit (Life Technologies). All genes examined were normalized to a housekeeping gene encoding *GAPDH*. Relative expression values were calculated from Ct values using the ΔΔ_CT_ method with freshly isolated hepatocytes as a control. Fold induction was calculated according to the formula 2^(Rt−Et)^/2^(Rn−En)^ [42]. PCRs were performed as follows: denaturation for 10 min at 95 °C, amplification with 40 cycles and 15 s at 95 °C, 40 s at a primer specific annealing temperature (as shown in Table 3), and 15 s at 72 °C. Each sample was set up in triplicates, and the experiment was repeated at least twice. Statistical significance of difference in target genes expression level between different treatments was assessed by One-way ANOVA.

### 4.6. CYP Activity Measurement

CYP enzyme activities of CYP2B6, CYP2D6, CYP2C9, and CYP3A4, were measured, as described. Briefly, the chosen substrates, the selected concentrations, the incubation times, and the measured metabolites are summarized in Table 4 Methanol, which was the initial solvent of the CYP substrates, was removed before use by evaporation, and the substrates were dissolved in culture medium. The cells were incubated with 100 µL of the respective reaction solution. After the described incubation times, the supernatants were removed and frozen at −80 °C until measurement. The enzymatic activity was measured by the company Pharmacelsus, Saarbrücken, Germany using a LC/MS based methodology [17,43].

### 4.7. SRB Staining for Normalisation of the Results

For normalization of the result from the CYP activity measurement, a Sulforhodamine B (SRB) staining was performed as described by Skehan et al. [44]. Therefore, cells were fixed to culture plastic with ice cold fixation buffer (95% Ethanol, 100 μL/well). The cells were incubated for at least 1 h at −20 °C. Then fixation buffer was removed, and cells were washed with H_2_O. Fixed cells were stained for 30 min (dark, RT) with 0.4% SRB dissolved in 1% acetic acid (100 μL/well). Then, SRB solution was removed and cells were washed three times with 1% acetic acid in order to wash out unbound dye. Plates were air dried and bound SRB was solubilized with 10 mM un-buffered TRIS (pH = 10.5; 100 μL/well) for 10–15 min on a shaker (RT, in the dark). The OD at 565 nm (SRB) and 690 nm (impurities) were measured with OMEGA plate reader (BMG Labtech, Ortenberg, Germany).

### 4.8. Statistic Analysis 

Statistical significance of differences between two groups was evaluated by non-parametric Mann–Whitney U-test. For comparison of the differences between more than two groups non-parametric Kruskal-Wallis H-test followed by Dunn’s multiple comparison test was performed using GraphPad Prism 5.00 Software, San Diego, CA, USA. Data are represented as means  ±  SEM of at least three independent experiments (*N* ≥ 3). All statistical comparisons were performed two-sided in the sense of an exploratory data analysis using *p* < 0.05 (*), *p* < 0.01 (**), and *p* < 0.001 (***) as level of significance.

## 5. Conclusions

Our results show that the epigenetic status of the hepatoblastoma cell line HepG2 shows the highest similarity with PHH compared to the other tested liver cancer cell lines (Huh7, HLE and AKN1). Also a shift of the epigenetic status of HepG2 cells, by stimulation with the epigenetic modifiers 5-AZA plus Vitamin C towards a profile characteristic for PHH could be achieved. Although these modifications lead to a reduction in the expression of the EMT marker gene *SNAIL* and induction in the expression of epithelial genes, but not to a significant increase in the expression and activity of CYP enzymes. As a possible reason, existing epigenetic modifications could be identified, which are deregulated in all four tested hepatoma cell lines as well as in long-term cultured hepatocytes. These genes are not linked yet to the metabolic activity of liver cells. Our results suggest that further epigenetic key players, which were identified in this study, are responsible for the hepatic differentiation as well as for the activity of the CYP enzymes. Changing the expression of these genes may not only be an approach to improving the metabolic activity of the cells, but may be additionally potential targets for tumor therapy.

## Figures and Tables

**Figure 1 ijms-20-00347-f001:**
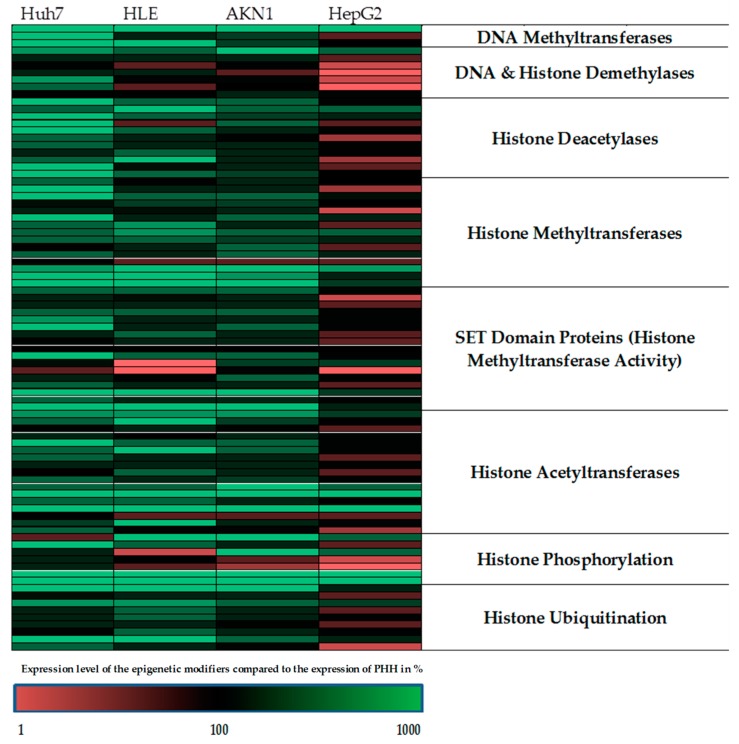
Summary of Human Epigenetic Chromatin Modification Enzymes PCR Array. The expression of 84 chromatin modification genes of four different hepatic cell lines in comparison with fresh isolated primary human hepatocytes (PHH) was shown as a heat map. The green color shows an upregulation of epigenetic modifier genes compared to PHH whereas the red color shows a downregulation of the corresponding genes compared to PHH.

**Figure 2 ijms-20-00347-f002:**
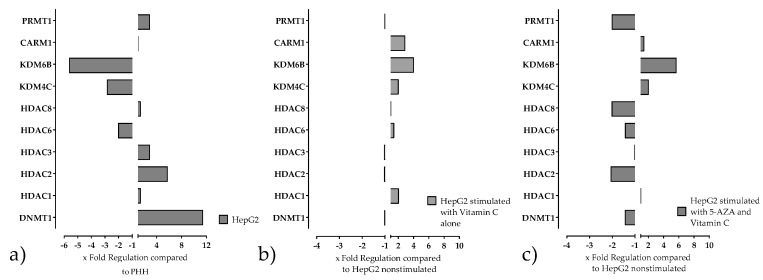
Expression of chromatin modifying genes in HepG2 cells before and after stimulation with Vitamin C alone or in combination with 5-Azacytidine (5-AZA) was measured by using the Chromatin Modification Array. Genes responsible for the maintenance of metabolic properties of primary human hepatocytes (PHH) are selected. (**a**) Expression of chromatin modification genes in HepG2 compared to PHH (**b**) Changes in the expression of chromatin modification genes in HepG2 caused by treatment with Vitamin C compared to non- stimulated HepG2 cells. (**c**) Changes in the expression of chromatin modification genes in HepG2 caused by 5-AZA in combination with Vitamin C compared to non- stimulated HepG2 cells.

**Figure 3 ijms-20-00347-f003:**
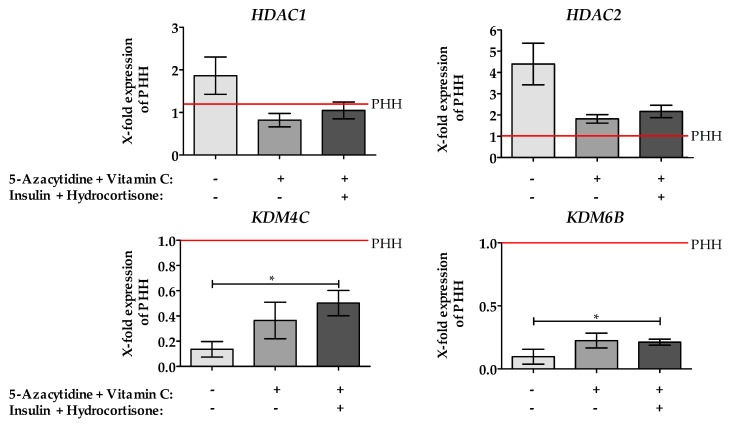
Effect of epigenetic modifying compounds on the expression of Chromatin Modifying enzymes. The expression levels of different chromatin modifying enzymes were measured by qRT-PCR. Unstimulated HepG2 cells (grey bars), as well as HepG2 cells after 48 h stimulation with 10 µM 5-AZA plus 0.5 mM Vitamin C with (dark bars)/without (black bars) Insulin and hydrocortisone, were used. As a control, freshly isolated hepatocytes were used, (red line). Values were normalized to *GAPDH* and represent the mean of *N* = 3, *n* = 3. For the positive control a pooled sample of five different primary human hepatocyte (PHH) donors was used. Bars represent mean ± SEM * *p* ≤ 0.05 as indicated.

**Figure 4 ijms-20-00347-f004:**
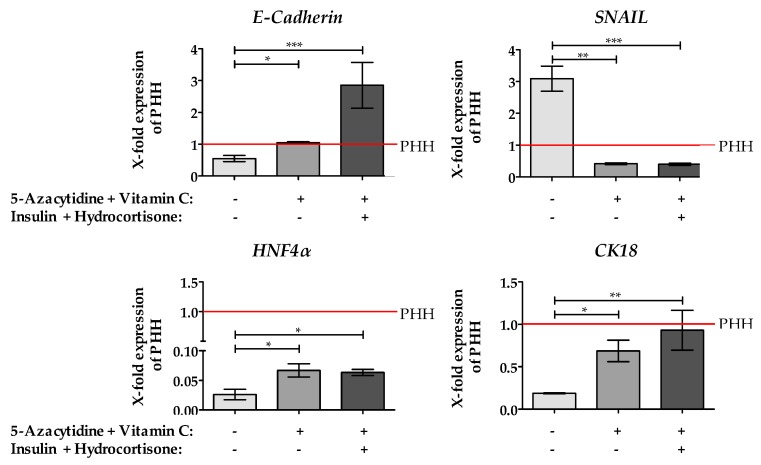
Effect of epigenetic modifying compounds on the expression of epithelial and EMT marker genes. Treatment of the cells with 10 µM 5-Azacytidine (5-AZA) plus 0.5 mM Vitamin C dark bars) and in the presence of insulin plus hydrocortisone (black bars) resulted in significant changes of epithelial and EMT marker genes. The expression level of the EMT marker Snail1 (*SNAIL*), the epithelial marker genes *E- Cadherin* and Cytokeratin 18 (*CK18*) and the hepatic key regulator gene Hepatocyte nuclear factor 4α (*HNF4α*) were measured by qRT-PCR. Unstimulated HepG2 cells (grey bars) as well as PHH (red line) served as control and reference culture. Values were normalized to *GAPDH* and represent the mean of *N* = 3, *n* = 3. As positive reference served a pool of five different primary human hepatocyte (PHH) donors. Bars represent Mean ± SEM, * *p* ≤ 0.05, ** *p* ≤ 0.01, *** *p* ≤ 0.001 as indicated.

**Figure 5 ijms-20-00347-f005:**
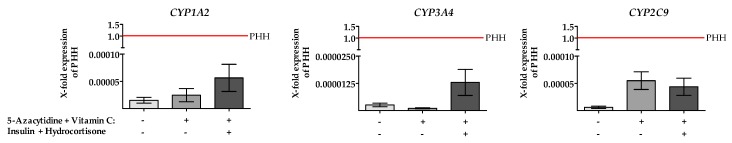
Incubation of HepG2 cells with 10 µM 5-Azacytidine plus 0.5 mM Vitamin C (dark bars) and in the presence of insulin plus hydrocortisone (black bars) increased basal Cytochrome P450 (CYP) gene expression. The gene expression of *CYP1A2*, *3A4* and *CYP2C9* were measured by qRT-PCR after 48 h of incubation. Unstimulated controls (grey bars) as well as primary human hepatocytes (PHH) (red line) were also cultured for 48 h. The values were normalized to GAPDH. Data represent the mean of *N* = 3, *n* = 3. As positive reference served a pool of five different PHH donors.

**Figure 6 ijms-20-00347-f006:**
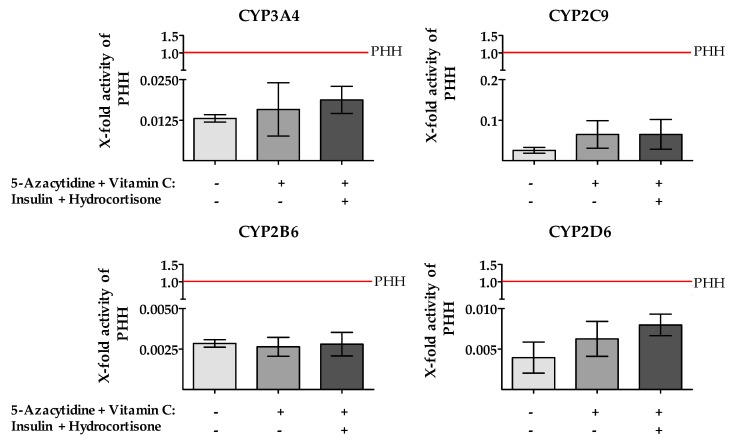
Activities of Cytochrome P450 (CYP) enzymes involved in drug metabolism are slightly increased after treatment with 5-Azacytidine (5-AZA) and Vitamin C. Supernatants were collected after incubation with specific substrates as indicated in materials and methods (4.4.6). Unstimulated HepG2 cells (grey bars) as well as HepG2 cells after 48 h stimulation with 10 µM 5-AZA plus 0.5 mM Vitamin C with (dark bars)/without (black bars) insulin and hydrocortisone were used. For control, freshly isolated primary human hepatocytes (PHH) from three donors were used, shown as red line. Bars represent mean ± SEM of three independent experiments; supernatants of at least three wells were pooled.

**Figure 7 ijms-20-00347-f007:**
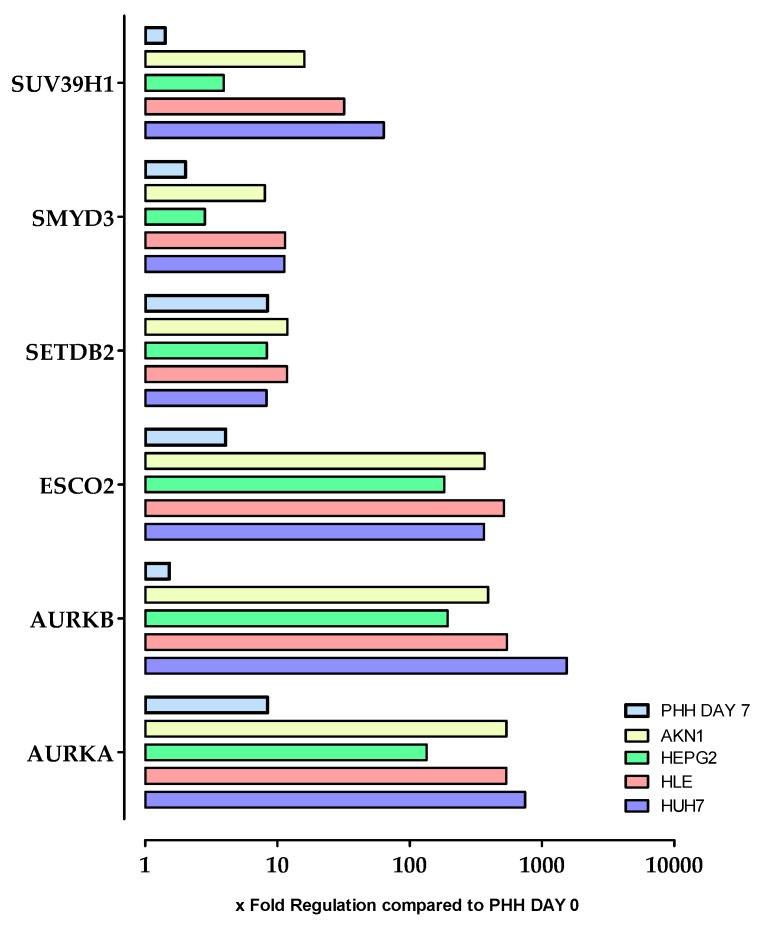
Chromatin modifying enzymes which are dramatically de-regulated in all tested cell lines and also de-regulated during primary human hepatocyte (PHH) de-differentiation in culture. Genes which are significantly de-regulated were selected. Expression of Chromatin Modification genes in hepatoma cell lines and PHH (day 7 after plating) were measured by using the Chromatin Modification Array.

**Table 1 ijms-20-00347-t001:** Description of the used cell lines.

Cell Line	Origin/Disease	Donor	Reference
HepG2	hepatoblastoma	15 year old Caucasian male	[39]
Huh7	HCC	57 year old Japanese male	[40]
HLE	HCC	68-year-old patient	[41]
AKN1	Healthy	10 year old male	[16]

**Table 2 ijms-20-00347-t002:** Compounds of stimulation medium, and timeline of stimulation.

Supplement	Concentration	Day	Unstimulated	Condition 1	Condition 2
**FCS**	10%	0–3	+	+	+
**P/S**	1%	0–3	+	+	+
**5-Azacytidin**	10 µM	1–3	-	+	+
**Vitamin C**	0.5 mM	1–3	-	+	+
**Human Insulin**	1 mM	2–3	-	-	+
**Hydrocortisone**	0.8 µg/mL	2–3	-	-	+

**Table 3 ijms-20-00347-t003:** Primers which were used for qRT-PCR.

Gen	Forward/Reverse Sequences	Annealing Tm	Product Length (bp)	GenBank Accession
*hHNF4A*	CAGGCTCAAGAAATGCTTCC GGCTGCTGTCCTCATAGCTT	59	101	NM_001287184.1
*hCK18*	GAGTATGAGGCCCTGCTGAACAT GCGGGTGGTGGTCTTTTGGAT	65	150	NM_199187.1
*hHDAC1*	AACTGCTAAAGTATCACCAGAGGGT CCGGTCCGTGGTGTAGAAGG	62	92	NM_004964.2
*hHDAC2*	TGAAGGAGAAGGAGGTCGAA GGATTTATCTTCTTCCTTAACGTCTG	59	124	NM_001527.3
*hCYP1A2*	GCTTCGGACAGCACTTCCCT AGAAGTCCAGGGGGTTCCCG	63	105	NM_000761.4
*hCYP3A4*	AGCCCAGCAAAGAGCAACAC TCCATATAGATAGAGGAGCACCAGG	60	147	NM_017460.5
*hCYP2C9*	GACATGAACAACCCTCAGGACTTT TGCTTGTCGTCTCTGTCCCA	62	145	NM_000771.3
*hKDM4C*	TGGATCCCAGATAGCAATGA TGTCTTCAAATCGCATGTCA	59	110	NM_001304340.1
*hKDM6B*	GGAGGCCACACGCTGCTAC GCCAGTATGAAAGTTCCAGAGCTG	63	112	NM_001348716.1
*hSNAIL*	ACCACTATGCCGCGCTCTT GGTCGTAGGGCTGCTGGAA	60	115	NM_005985.3
*hGAPDH*	TGCACCACCAACTGCTTAGC GGCATGGACTGTGGTCATGAG	59	87	NM_002046.3

**Table 4 ijms-20-00347-t004:** Substrates, concentrations, conditions and measured reactions of the CYP activity measurement.

Substrate	Isoenzyme	Incubation Time in Hours	Concentration	Reaction
Bupropion	CYP2B6	1	100 μM	Bupropion-hydroxylation
Diclofenac	CYP2C9	1	9 μM	Diclofenac-4’-hydroxylation
Testosterone	CYP3A4	1	50 μM	Testosterone-6β-hydroxylation
Bufuralol	CYP2D6	2	9 μM	Bufuralol-1-hydroxylation

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
