# Peer review of "Epigenetic Modifications of the Liver Tumor Cell Line HepG2 Increase Their Drug Metabolic Capacity"

_ijms, 2019, doi:10.3390/ijms20020347_

Reviewer 1 Report

The paper of Ruoss et al describes efforts to « reactivate » the drug metabolism (Phase I) capacities of cancerous hepatic cell lines to the level of freshly isolated liver cells by mean of pharmacological treatments (vit C and Aza plus, later on, insulin and hydrocortisone.

It is clear to me that those experiments are important and should be continued until we find a way to boost the HepG2 cell line – or any other one cell line).

The paper is clear, with some strange sentences at some points, such as line 149: “.. improved the expression E-Cadherin more than threefold compared to PHH”. ‘Improve” is inadequate. Enhance? But those are minor and could be fixed rapidly.

From the most formal point of view, further details on the origin of the liver samples should be given. The way they were obtained, and from where.

I wondered what a LC-HPLC/MS technology is. It is not described at all, and only refereed to another paper. Please, at least give some details.  LC-HPLC/MS seems to refer to a HPLC/MS technique, unless this is a 2 dimensional LC. But then which ones and why LC followed by HPLC?

Another “minor” point: are Vit-C, hydrocortisone, insulin and Aza inhibitors of the Cyp measured activities? This is should be checked before claiming anything else. I also would like to see a WB of the Cyp’s once the HepG2 were treated with the Vit-C and Aza, that also would tell us something on what is actually happening. I guess there are specific Ab (not sure, though)???

Author Response

Dear Editor, dear Reviewers,

We like to thank the editor and reviewers for giving us the opportunity to re-revise our manuscript. We appreciated your comments and have addressed all issues regarding your recommendations and edited the manuscript accordingly. As recommended we included one additional table into the manuscript and elaborated on other concerns. The detailed answers to your questions are summarized below. All changes performed in the manuscript are highlighted in yellow.

We would like to resubmit our revised manuscript to the International Journal of Molecular Sciences and hope that the revised manuscript meets now the high standards of this journal.

All authors have approved the final submitted manuscript and no conflict of interest exists between the authors

Yours sincerely,

Prof. Dr. Andreas K. Nüssler

Editor:

We appreciated your comments and have addressed all issues regarding your recommendations and edited the manuscript accordingly. As recommended, we have changed the title by specifying that the manuscript deals with drug metabolism rather than a paper regarding cancer metabolism. In addition, we agree that based on our data no statement can be made about the actual EMT status of the cell. Moreover, we included chromatin array data from HepG2 cells which were stimulated with Vitamin C alone to the manuscript. The data presented and the already earlier published results clearly showed, that the combination of 5-AZA and Vitamin C is most efficient to stop cell growth with the lowest effect of toxicity [1]. The addition of hydrocortisone and insulin was not used as a ‘classical treatment’. According to the literature HepG2 cells are usually cultured without cortisone and insulin [2]. The purpose here was to increase the metabolic activity of the cancer cell line by adding these components, since they are part of established human hepatocyte cultures [3].

Reviewer 1:

Point 1: The paper is clear, with some strange sentences at some points, such as line 149: “.. improved the expression E-Cadherin more than threefold compared to PHH”. ‘Improve” is inadequate. Enhance? But those are minor and could be fixed rapidly.

Response 1:  Thank you for this comment, we change the addressed term and re-checked the manuscript again regarding other strange sentences

Point 2: From the most formal point of view, further details on the origin of the liver samples should be given. The way they were obtained, and from where.

Response 2: Thank you for this comment. The hepatocytes were isolated from liver capsules, which were collected at the clinic of General, Visceral and Transplant Surgery (Tübingen) and the clinic of Hepatobiliary Surgery and Visceral Transplantation (Leipzig) in the context of primary and secondary liver surgery; only tumor free tissue was isolated as described in earlier publications [4,5]. This description has been modified accordingly.

Point 3: I wondered what a LC-HPLC/MS technology is. It is not described at all, and only refereed to another paper. Please, at least give some details.  LC-HPLC/MS seems to refer to a HPLC/MS technique, unless this is a 2 dimensional LC. But then which ones and why LC followed by HPLC?

Response 3: Please excuse the confusion. An LC/ MS based method was used. Because of a changed drug legislation in Germany we used in contrast to the shown reference testosterone instead of midazolam for the measurement of CYP3A4 activity, therefore we include a further reference [3] into the recent version of the manuscript.

Point 4: Another “minor” point: are Vit-C, hydrocortisone, insulin and Aza inhibitors of the Cyp measured activities? This is should be checked before claiming anything else. I also would like to see a WB of the Cyp’s once the HepG2 were treated with the Vit-C and Aza, that also would tell us something on what is actually happening. I guess there are specific Ab (not sure, though)???

Response 4: Thanks for this interesting question, based on the literature, there is no evidence that the substances used in the concentration used have a negative effect on the expression or activity of CYP enzymes [6,7]. Additionally, it should also be taken into account that hydrocortisone and insulin were not used as a classical treatment, but rather a physiological concentration was used here, which is also used for example for the cultivation of human hepatocytes [3]. A washing step between incubation with the substances and the activity measurement with PBS also ensured that all conditions were comparable.

At the beginning of this study it was planned to measure the CYP enzymes fluorescence-based and additionally to determine the protein expression of the CYP via WB, but according to literature review [8,9] we have strayed from it. The reason for this is that with both methods it is not possible to differentiate correctly between different CYP isoforms which is especially important in cancer cells and which express the mature as well as the embryonal CYP isoform [9,10], since we have not the possibility to distinguish between CYP isoforms by mass spectrometry as shown by Redlich et al. [9] we decide to measure the specific CYP isoform metabolites by LC/ MS instead of performing a WB which is more precise than a WB.

1.         Sajadian, S.O.; Tripura, C.; Samani, F.S.; Ruoss, M.; Dooley, S.; Baharvand, H.; Nussler, A.K. Vitamin C enhances epigenetic modifications induced by 5-azacytidine and cell cycle arrest in the hepatocellular carcinoma cell lines HLE and Huh7. Clin Epigenetics 2016, 8, 46, doi:10.1186/s13148-016-0213-6.

2.         Lin, J.; Schyschka, L.; Mühl-Benninghaus, R.; Neumann, J.; Hao, L.; Nussler, N.; Dooley, S.; Liu, L.; Stöckle, U.; Nussler, A.K., et al. Comparative analysis of phase I and II enzyme activities in 5 hepatic cell lines identifies Huh-7 and HCC-T cells with the highest potential to study drug metabolism. Arch Toxicol 2012, 86, 87-95, doi:10.1007/s00204-011-0733-y.

3.         Ruoß, M.; Häussling, V.; Schügner, F.; Olde Damink, L.; Lee, S.; Ge, L.; Ehnert, S.; Nussler, A. A Standardized Collagen-Based Scaffold Improves Human Hepatocyte Shipment and Allows Metabolic Studies over 10 Days. Bioengineering 2018, 5, 86.

4.         Pfeiffer, E.; Kegel, V.; Zeilinger, K.; Hengstler, J.G.; Nussler, A.K.; Seehofer, D.; Damm, G. Featured Article: Isolation, characterization, and cultivation of human hepatocytes and non-parenchymal liver cells. Exp Biol Med (Maywood) 2015, 240, 645-656, doi:10.1177/1535370214558025.

5.         Knobeloch, D.; Ehnert, S.; Schyschka, L.; Buchler, P.; Schoenberg, M.; Kleeff, J.; Thasler, W.E.; Nussler, N.C.; Godoy, P.; Hengstler, J., et al. Human hepatocytes: isolation, culture, and quality procedures. Methods in molecular biology (Clifton, N.J.) 2012, 806, 99-120, doi:10.1007/978-1-61779-367-7_8.

6.         Heeswijk, R.P.G.; Cooper, C.L.; Foster, B.C.; Chauhan, B.M.; Shirazi, F.; Seguin, I.; Phillips, E.J.; Mills, E. Effect of High‐Dose Vitamin C on Hepatic Cytochrome P450 3A4 Activity. Pharmacotherapy: The Journal of Human Pharmacology and Drug Therapy 2005, 25, 1725-1728, doi:10.1592/phco.2005.25.12.1725.

7.         Chen, Y.; Liu, L.; Laille, E.; Kumar, G.; Surapaneni, S. In vitro assessment of cytochrome P450 inhibition and induction potential of azacitidine. Cancer chemotherapy and pharmacology 2010, 65, 995-1000, doi:10.1007/s00280-010-1245-9.

8.         Ung, Y.; Ong, C.; Pan, Y. Current High-Throughput Approaches of Screening Modulatory Effects of Xenobiotics on Cytochrome P450 (CYP) Enzymes. High-Throughput 2018, 7, 29.

9.         Redlich, G.; Zanger, U.M.; Riedmaier, S.; Bache, N.; Giessing, A.B.M.; Eisenacher, M.; Stephan, C.; Meyer, H.E.; Jensen, O.N.; Marcus, K. Distinction between Human Cytochrome P450 (CYP) Isoforms and Identification of New Phosphorylation Sites by Mass Spectrometry. Journal of Proteome Research 2008, 7, 4678-4688, doi:10.1021/pr800231w.

10.       Rodrigues, A.D. Drug-drug interactions; CRC Press: 2001.

Reviewer 2 Report

Manuscript (Ref #. IJMS-421474-review-v1)

Title: Epigenetic modifications of the liver tumor cell line HepG2 increase their metabolic capacity for in vitro testing

Corresponding author: Professor S. Sajadian

In general, the data in present studies are good and support the major conclusions of this manuscript. However, following issues need to be considered prior to considering the manuscript of publication.

Specific comments are as follows:

1. English: The manuscript has numerous typographical errors that should be corrected. Authors use capital letters indiscriminately. Authors need to be careful in writing manuscript. The entire manuscript should be rewritten by the expert in correct English.

2. Authors used four different cell lines such as Huh7, HLE, HepG2, and AKN1. The authors need to describe these four cell lines in a certain order, if possible.

3. Materials and Methods section:

1) When they name a certain chemical company, they should provide proper information of company name, location such as city and/or state (abbreviated), and country. If you once named a company with its information, you must write a company name only thereafter.

2) Measurements must be written with a certain consistency. Authors use three different types of hour(s) such as 48 hrs, 48 hr, or 48 h; Temperature: 95°C or 95 °C.

3) Typos: Life technologies should be Life Technologies at Line 373.

4. Discussion: Please discuss the physiological relevance and potential clinical applicability of its finding.

5. Conclusions: Please rewrite the first sentence. Authors are trying to say that HepG2 cell line is better than other cell lines such as Huh7, HLE and AKN-1. However, the current sentence are including HepG2 cell line among other cell lines.

6. Reference section: Author should consult and peruse carefully recent issues of the journal, International Journal of Molecular Sciences, for format and style.

7. Authors should always keep the same name when naming it. However, there are several cases. Examples: AKN-1 at Line 408 or AKN1 at Line 76 and Table 1; SNAIL or Snail at Figure 4; HepG2 cells (hepatoma cell line or hepatoblastoma cell line at Line 147 and Line 238; PRMT 1 at Line 249 or PRMT1 at Lin3 246; CK 18 at Figure 4 or CK18 at Line 279; PHH or pHH at Figure 4; CYP 1A2 or CYP1A2 at Table 2; vitamin C or Vitamin C at everywhere.

8. Typos: figure 5 at Line 172; SuppTab.2 at Line 214, etc.

9. It is really hard to understand the following sentence: “Data represent an average of N = 3, n = 3.” At Figure 3; “0.08 mio cells/cm2…” at Line 334.

10. Table 3: Please double check the reaction type of testosterone.

Overall, the manuscript can be considered to be accepted with major revision as indicated above.

Author Response

Dear Editor, dear Reviewers,

We like to thank the editor and reviewers for giving us the opportunity to re-revise our manuscript. We appreciated your comments and have addressed all issues regarding your recommendations and edited the manuscript accordingly. As recommended we included one additional table into the manuscript and elaborated on other concerns. The detailed answers to your questions are summarized below. All changes performed in the manuscript are highlighted in yellow.

We would like to resubmit our revised manuscript to the International Journal of Molecular Sciences and hope that the revised manuscript meets now the high standards of this journal.

All authors have approved the final submitted manuscript and no conflict of interest exists between the authors

Yours sincerely,

Prof. Dr. Andreas K. Nüssler

Editor:

We appreciated your comments and have addressed all issues regarding your recommendations and edited the manuscript accordingly. As recommended, we have changed the title by specifying that the manuscript deals with drug metabolism rather than a paper regarding cancer metabolism. In addition, we agree that based on our data no statement can be made about the actual EMT status of the cell. Moreover, we included chromatin array data from HepG2 cells which were stimulated with Vitamin C alone to the manuscript. The data presented and the already earlier published results clearly showed, that the combination of 5-AZA and Vitamin C is most efficient to stop cell growth with the lowest effect of toxicity [1]. The addition of hydrocortisone and insulin was not used as a ‘classical treatment’. According to the literature HepG2 cells are usually cultured without cortisone and insulin [2]. The purpose here was to increase the metabolic activity of the cancer cell line by adding these components, since they are part of established human hepatocyte cultures [3].

Reviewer 2:

Point 1: English: The manuscript has numerous typographical errors that should be corrected. Authors use capital letters indiscriminately. Authors need to be careful in writing manuscript. The entire manuscript should be rewritten by the expert in correct English.

Response 1: Thanks for the comment, a native speaker had re-checked the manuscript and we corrected the typographical errors

Point 2: Authors used four different cell lines such as Huh7, HLE, HepG2, and AKN1. The authors need to describe these four cell lines in a certain order, if possible.

Response 2: Thanks for this comment, we have added a new table in which the cell lines used are clearly and uniformly shown.

Point 3: Materials and Methods section:

1) When they name a certain chemical company, they should provide proper information of company name, location such as city and/or state (abbreviated), and country. If you once named a company with its information, you must write a company name only thereafter.

2) Measurements must be written with a certain consistency. Authors use three different types of hour(s) such as 48 hrs, 48 hr, or 48 h; Temperature: 95°C or 95 °C.

3) Typos: Life technologies should be Life Technologies at Line 373.

Response 3: Please excuse certain inconsistencies, we have thoroughly revised the manuscript accordingly.

Point 4: Discussion: Please discuss the physiological relevance and potential clinical applicability of its finding.

Response 4: Thanks for this comment, in the new version we have addressed the possibilities but also the limitations of using epigenetically altering substances like 5-AZA in HCC therapy and discussed possible targets to further epigenetically reactivate the cells for possible therapy as well as for the use as a predictive in vitro model.

Point 5: Conclusions: Please rewrite the first sentence. Authors are trying to say that HepG2 cell line is better than other cell lines such as Huh7, HLE and AKN-1. However, the current sentence are including HepG2 cell line among other cell lines.

Response 5: Thanks for that comment we have revised the mentioned sentence accordingly.

Point 6: Reference section: Author should consult and peruse carefully recent issues of the journal, International Journal of Molecular Sciences, for format and style.

Response 6: Thanks for this comment we have revised the references according to the recommendations of the MDPI template.

Point 7: Authors should always keep the same name when naming it. However, there are several cases. Examples: AKN-1 at Line 408 or AKN1 at Line 76 and Table 1; SNAIL or Snail at Figure 4; HepG2 cells (hepatoma cell line or hepatoblastoma cell line at Line 147 and Line 238; PRMT 1 at Line 249 or PRMT1 at Lin3 246; CK 18 at Figure 4 or CK18 at Line 279; PHH or pHH at Figure 4; CYP 1A2 or CYP1A2 at Table 2; vitamin C or Vitamin C at everywhere

Response 7: Please excuse that there were many inconsistencies in the first version of the manuscript, we have carefully revised it

Point 8: Typos: figure 5 at Line 172; SuppTab.2 at Line 214, etc.

Response 8: We have corrected the typos you have mentioned, during this process, we noticed that the supplementary material is more like Figures and not like Tables, which we have changed accordingly.

Point 9: It is really hard to understand the following sentence: “Data represent an average of N = 3, n = 3.” At Figure 3; “0.08 mio cells/cm2…” at Line 334.

Response 9:  We have changed the first sentence that you mentioned accordingly. We also have revised the description of the cell numbers used to this format: 0.8 x 104 cells/cm2, additionally the description now corresponds to the form we have also chosen for the description of the used PHH.

Point 10: Table 3: Please double check the reaction type of testosterone.

Response 10: Thanks for the comment, we checked the reaction type again and but we found no mistake, the testosterone 6β-hydroxylation is the prototypic reaction

of cytochrome P450 3A4 [4]

1.         Sajadian, S.O.; Tripura, C.; Samani, F.S.; Ruoss, M.; Dooley, S.; Baharvand, H.; Nussler, A.K. Vitamin C enhances epigenetic modifications induced by 5-azacytidine and cell cycle arrest in the hepatocellular carcinoma cell lines HLE and Huh7. Clin Epigenetics 2016, 8, 46, doi:10.1186/s13148-016-0213-6.

2.         Lin, J.; Schyschka, L.; Mühl-Benninghaus, R.; Neumann, J.; Hao, L.; Nussler, N.; Dooley, S.; Liu, L.; Stöckle, U.; Nussler, A.K., et al. Comparative analysis of phase I and II enzyme activities in 5 hepatic cell lines identifies Huh-7 and HCC-T cells with the highest potential to study drug metabolism. Arch Toxicol 2012, 86, 87-95, doi:10.1007/s00204-011-0733-y.

3.         Ruoß, M.; Häussling, V.; Schügner, F.; Olde Damink, L.; Lee, S.; Ge, L.; Ehnert, S.; Nussler, A. A Standardized Collagen-Based Scaffold Improves Human Hepatocyte Shipment and Allows Metabolic Studies over 10 Days. Bioengineering 2018, 5, 86.

4.         Krauser, J.A.; Guengerich, F.P. Cytochrome P450 3A4-catalyzed testosterone 6β-hydroxylation stereochemistry, kinetic deuterium isotope effects, and rate-limiting steps. Journal of Biological Chemistry 2005, 280, 19496-19506.

Round  2

Reviewer 2 Report

Manuscript (Ref #. IJMS-421474-review-v2)

Title: Epigenetic modifications of the liver tumor cell line HepG2 increase their drug metabolic capacity

Corresponding author: Professor S. Sajadian

Even though authors revised the manuscript, there are still some unsolved issues.

Specific comments are as follows:

Line 356: 0.8 x 104 cell/cm2 should be 8 x 103 cell/cm2.

Table 1: Concentration of Vitamin C should be 0.5 mM, not 0,5 mM.

Figure legend of Figure 1: HuH7 should be Huh7.

Overall, the manuscript can be considered to be accepted with minor revision as indicated above.

Author Response

Dear Editor, dear Reviewer,

We like to thank the editor and reviewers for giving us the opportunity to re-revise our manuscript. We appreciated your comments and have addressed all issues regarding your recommendations and edited the manuscript accordingly. All changes of the minor revision performed in the manuscript are highlighted in green.

We would like to resubmit our revised manuscript to the International Journal of Molecular Sciences and hope that the revised manuscript meets now the high standards of this journal.

All authors have approved the final submitted manuscript and no conflict of interest exists between the authors

Yours sincerely,

Prof. Dr. Andreas K. Nüssler